# Factors associated with prolonged hospitalizations for COVID-19 during the first three waves of the pandemic: Evidence from a Southeastern State of Brazil

Juliana Rodrigues Tovar Garbin[1,2], Franciéle Marabotti Costa Leite[2], Ana Paula Brioschi dos Santos[2,3,4], Larissa Soares Dell'Antonio[3,4], Cristiano Soares da Silva Dell'Antonio[3,5], Luís Carlos Lopes-Júnior[2]*

1 Instituto Federal do Espírito Santo/IFES, Espírito Santo, Brazil, 2 Graduate Program in Public Health - PPGSC/UFES, Vitória, Espírito Santo, Brazil, 3 Secretaria de Estado da Saúde do Espírito Santo, Ministry of Health, Vitória, Espírito Santo, Brazil, 4 Instituto Capixaba de Ensino, Pesquisa e Inovação - ICEPi, Vitória, Espírito Santo, Brazil, 5 Hospital Sírio Libanês, São Paulo, Brazil

* lopesjr.lc@gmail.com

## Abstract

A comprehensive understanding of the factors influencing the epidemiological dynamics of COVID-19 across the pandemic waves—particularly in terms of disease severity and mortality—is critical for optimizing healthcare services and prioritizing high-risk populations. Here we aim to analyze the factors associated with short-term and prolonged hospitalization for COVID-19 during the first three pandemic waves. We conducted a retrospective observational study using data from individuals reported in the e-SUS-VS system who were hospitalized for COVID-19 in a state in a southeast state of Brazil. Hospitalization duration was classified as short or prolonged based on a 7-day cutoff, corresponding to the median length of hospital stay during the second pandemic wave. Bivariate analyses were performed using the chi-square test for heterogeneity. Logistic regression models were used to estimate odds ratios (ORs) and their respective 95% confidence intervals (CIs), with statistical significance set at 5%. When analyzing hospitalization duration across the three waves, we found that 51.1% (95%CI: 49.3–53) of hospitalizations in the first wave were prolonged. In contrast, short-duration hospitalizations predominated in the second (54.7%; 95% CI: 52.4–57.0) and third (51.7%; 95% CI: 50.2–53.2) waves. Factors associated with prolonged hospitalization varied by wave. During the first wave, older adults (≥60 years) (OR=1.67; 95%CI: 1.35–2.06), individuals with ≥10 symptoms (OR=2.03; 95%CI: 1.04–3.94), obese individuals (OR=2.0; 95%CI: 1.53–2.74), and those with ≥2 comorbidities (OR=2.22; 95%CI: 1.71–2.89) were more likely to experience prolonged hospitalization. In the second wave, he likelihood of extended hospital stays was higher among individuals aged ≥60 years (OR=2.04; 95%CI: 1.58–2.62) and those with ≥2 comorbidities (OR=1.77; 95%CI: 1.29–2.41). In the third wave, prolonged

**Data availability statement:** All relevant data are within the paper and its Supporting Information files.

**Funding:** The author(s) received no specific funding for this work.

**Competing interests:** The authors have declared that no competing interests exist.

hospitalization was more frequent among older adults (OR=1.89; 95%CI: 1.65–2.17,), individuals with 5–9 symptoms (OR=1.52; 95%CI: 1.20–1.92), obese individuals (OR=2.2; 95%CI: 1.78–2.73), and those with comorbidities (OR=1.45; 95%CI: 1.22–1.72 and OR=2.0; 95%CI: 1.69–2.45). In conclusion, we identified variations in hospitalization patterns across the pandemic waves, although the differences were relatively subtle. These variations likely reflect gradual shifts in the risk factors associated with prolonged hospital stays. Our findings highlight t the importance of implementing targeted public health interventions, particularly those designed to reduce disease severity and improve clinical outcomes among vulnerable populations at greater risk of extended hospitalization.

## Introduction

Three years after the isolation and subsequent classification of the novel Coronavirus as SARS-CoV-2 [1], the World Health Organization (WHO) officially declared the end of the Public Health Emergency of International Concern (PHEIC) related to the COVID-19 pandemic on May 5, 2023. By May 10, 2023, a total of 765,903,278 confirmed cases and 6,927,378 deaths had been reported worldwide, with 13,350,530,518 vaccine doses administered [2]. During the same period, Brazil recorded 37,553,337 confirmed cases and 702,421 deaths, corresponding to a mortality rate of 1.9% [3]. Throughout the pandemic, SARS-CoV-2 has demonstrated its ability to adapt to the human host, primarily through the emergence of novel variants that enhance its transmissibility [4]. In Brazil, cases of COVID-19 have been reported across multiple viral lineages, including four variants of concern and two variants of interest, as classified by the World Health Organization (WHO) [5]. Consequently, the emergence of these new strains has been associated with shifts in symptom prevalence of symptoms, transmissibility, and disease-related complications [6].

Overall, studies indicate that approximately 80% of COVID-19 cases are asymptomatic, while the remaining 20% may progress to severe and critical forms requiring hospitalization [7,8]. A meta-analysis estimated the prevalence of severe cases at 17.84% and critical cases at 4.9% [9]. Among individuals diagnosed with COVID-19, the most frequently reported clinical symptoms include fever (83%), cough (60%), and dyspnea (42%), often in conjunction with underlying comorbidities such as hypertension, obesity, diabetes, and cardiovascular disease [6,8,10,11].

It is important to highlight that, patients with severe manifestations, such as acute respiratory distress syndrome (ARDS) and those requiring intensive care unit (ICU) admission, are more likely to develop long-term sequelae due to the persistence of symptoms, commonly referred to as long COVID-19 [12]. A comprehensive understanding of the factors driving epidemiological shifts in disease dynamics—particularly in terms of severity and mortality—across different pandemic waves can significantly improve the effectiveness of healthcare services in in addressing the needs of high-risk populations [8,11,13]. Brazil, one of the most socially unequal countries in the world, experienced significant regional disparities in the impact of the COVID-19

pandemic as the disease progressed. Following the introduction of SARS-CoV-2, a sharp increase in cases of Severe Acute Respiratory Syndrome (SARS) was recorded during the early epidemiological weeks of 2020. At this stage, structural inequalities became evident, particularly given that more than 90% of municipalities lacked the necessary resources to manage severe cases. This resource scarcity led to the collapse of the Brazilian healthcare system and, consequently, a surge in mortality rates. However, the implementation of vaccination proved to be an effective countermeasure, significantly reducing the incidence of severe cases and preventing death [14].

A comprehensive understanding of COVID-19, particularly regarding the clinical characteristics exhibited by affected individuals through rigorous case analyses, can facilitate the development and implementation of novel health interventions aimed at disease control [15]. Additionally, a detailed characterization of clinical patterns enables the early identification of complications, contributing to more effective resource allocation, the refinement of clinical protocols, and the strengthening of epidemiological surveillance—especially in high-demand healthcare settings, as observed during the pandemic.

Hence, the objective of this study was to analyze the factors associated with short-term and prolonged hospitalization for COVID-19 during the first three pandemic waves.

## Materials and methods

This study is a retrospective, descriptive investigation utilizing data from individuals who were reported and hospitalized for COVID-19 in the state of Espírito Santo (ES), located in the Southeast region of Brazil. Espírito Santo has a population of 4,108,508 residents across 78 municipalities [14]. The research protocol was approved by the Research Ethics Committee (approval number 5,180,941 granted on December 20, 2021). Data were obtained from the Espírito Santo State Department of Public Health (SESA-ES), via the e-SUS Health Surveillance information system (e-SUS-VS). The study included notifications of hospitalized patients between March 1, 2020 and July 31, 2021. The adopted inclusion criterion was a laboratory-confirmed diagnosis of COVID-19 by RT-PCR, with hospitalization in Espírito Santo specifically due to the disease. Exclusion criteria included hospitalizations occurring before symptom onset, absence of discharge or outcome date, and hospital stays shorter than 24 hours.

### Measures

In this study, the classification of short-term and prolonged hospitalizations across the three pandemic waves was based on 7-day cutoff, corresponding to the median hospital stay observed during the second wave. Hospitalizations lasting up to 7 days were classified as short-term, whereas those lasting 8 days or longer were categorized as long-term. Using the median length of stay as a threshold has been reported in the literature as a method to assess hospital course in homogeneous groups with distinct clinical outcomes [16]. Furthermore, the choice of a 7-day cutoff is consistent with national data indicating an average hospital stay of 6.9 days among patients admitted to public hospitals within the Brazilian Unified Health System (SUS), reinforcing the clinical representativeness of this threshold for our study population [17].

Hospitalizations in both general wards and intensive care units (ICUs) were taken into account. Sociodemographic characteristics considered in this study included age group (≤59 years; ≥60 years), gender (male; female), education attainment (<8 years; 8–11 years; ≥12 years of education).

For symptom assessment, the presence of one or more of the following symptoms was considered: fever, dyspnea, nasal flaring, intercostal indrawing, cyanosis, oxygen saturation below 95%, coma, cough, sputum production, nasal or conjunctival congestion, rhinorrhea, sore throat, difficulty swallowing, diarrhea, nausea/vomiting, headache, irritability/confusion, weakness, pharyngeal exudate, conjunctivitis, seizure, loss of smell (anosmia), and loss of taste (ageusia). The total number of symptoms experienced by each patient was recorded for analysis.

The comorbidity scoring system in this study was based on the presence of one or more of the following conditions: chronic lung disease, chronic cardiovascular disease, chronic kidney disease, chronic liver disease, diabetes mellitus, HIV

infection, neoplasms (cancer), history of bariatric surgery, tuberculosis, chronic neurological disease, or neuromuscular disease.

## Data analysis

Statistical analyses were performed using Stata version 15.0. Descriptive analysis included the calculation of absolute and relative frequencies along with their corresponding confidence intervals. Bivariate analysis was conducted using the chi-square test for heterogeneity. Subsequently, crude and adjusted odds ratios (OR) with their corresponding 95% confidence intervals were estimated using a logistic regression model. Variables were included in the regression model based on a p-value threshold of <0.20 to control for potential confounding factors, following a predefined conceptual model. These variables served as sole selection criteria for confounder adjustment. Additionally, supplementary analyses were conducted to assess the association between pandemic waves and hospitalization categorized as short versus prolonged, using robust standard errors to account for potential violations of homoscedasticity. A robust linear regression model was applied, with hospitalization duration in days as the continuous dependent variable and the pandemic wave as the independent variable. Statistical significance was set at 5%.

## Results

The distribution of hospitalizations varied across the three waves, with the highest number occurring during the third wave (48.9% or 4,410 hospitalizations), followed by the first wave (30.9% or 2,786 hospitalizations) and the second wave (20.2% or 1,821 hospitalizations). However, when examining the length of hospital stay, the first wave had the longest mean duration of 11.6 days (SD ± 12.75) and a median of 8 days. In contrast, the second wave had a shorter mean duration of 9.96 days (SD ± 10.09) and a median of 7 days. The third wave had a longer mean duration than the second wave but remained shorter than the first wave, with a mean of 11.35 days (±SD 12.34) and a median of 7 days. The differences observed were statistically significant (p < 0.001).

Furthermore, when analyzing the frequency of short-term and prolonged hospitalizations across the three waves, it was observed that during the first wave, 51.1% (95% CI: 49.3–53) of hospitalizations were classified as long-term. However, in the second and third waves, the majority of hospitalizations (54.7% and 51.7%; 95% CI: 52.4–57.0 and 50.2–53.2, respectively) were categorized as short-term, as shown in Table 1.

Regarding the sociodemographic and clinical characteristics, in the first wave, the majority of hospitalized patients were aged 60 or older (52.1%), male (54.7%), non-white (60.3%), and had less than 8 years of education (46.7%) (Table 2). Age group was associated with prolonged hospitalization (p < 0.001), with older adults (≥60 years) exhibiting a higher prevalence of extended hospital stays (58.4%). Most individuals presented with 2–4 symptoms (53.1%). Among risk factors and comorbidities, 5.1% of hospitalized patients were smokers, 12% were obese, and approximately 35% had 2 or more comorbidities. Smoking, obesity, and the number of comorbidities were associated with prolonged hospitalization (p < 0.05).

In the second wave, the majority of patients were also aged 60 or older (55.4%), male (53.8%), non-white (53.5%), and had 8–11 years of education (35.5%). Age group and years of education were associated with the duration of hospitalization for COVID-19. Individuals aged 60 or older and those with lower educational attainment had a higher occurrence of prolonged hospitalization (p < 0.001), as shown in Table 3.

**Table 1. COVID-19 hospitalizations in Espírito Santo, according to length of hospital stay in days, by waves, 2020-2021.**

| | Wave 1 | | Wave 2 | | Wave 3 | |
|---|---|---|---|---|---|---|
| Length of hospitalization | N (%) | 95% CI | N (%) | 95% CI | N (%) | 95% CI |
| Up to 7 days (short) | 1362 (48.9) | 47-50.7 | 996 (54.7) | 52.4-57.0 | 2280 (51.7) | 50.2-53.2 |
| 8 or more days (long) | 1424 (51.1) | 49.3-53 | 825 (45.3) | 43.0-47.6 | 2130 (48.3) | 46.8-49.8 |

**Table 2. Sociodemographic and clinical characterization of hospitalizations for COVID-19 according to duration in the 1st wave, Espírito Santo, 2020-2021.**

| Variables | Sample | | Short (Up to 7 days) | | Long (8 days or more) | | $x^2$* | p-value |
|---|---|---|---|---|---|---|---|---|
| | N (%) | 95% CI | N (%) | 95% CI | N (%) | 95% CI | | |
| **Sociodemographic characteristics** | | | | | | | | |
| **Age range (N = 2864)** | | | | | | | **45.94** | **<0.001** |
| Up to 59 years | 1333 (47.8) | 46.0–49.7 | 741(54.4) | 51.7–57.0 | 592 (41.6) | 39.0–44.2 | | |
| ≥60 | 1453 (52.2) | 50.3–54 | 621 (45.6) | 43.0–48.3 | 8323 (58.4) | 55.8–61.0 | | |
| **Gender (N = 2864)** | | | | | | | 4.09 | 0.043 |
| Male | 1263 (45.3) | 43.5–47.2 | 644 (47.3) | 44.6–49.9 | 619 (43.5) | 40.9–46.1 | | |
| Female | 1523 (54.7) | 52.8–56.5 | 718 (52.7) | 50.1–55.4 | 805 (56.5) | 53.9–59.1 | | |
| **Race/color (N = 2283)** | | | | | | | 0.86 | 0.355 |
| White | 881 (39.7) | 37.7–41.8 | 408 (38.7) | 35.8–41.7 | 473 (40.6) | 37.8–43.4 | | |
| Non-white | 1337 (60.3) | 58.2–62.3 | 646 (61.3) | 58.3–64.2 | 691 (49.4) | 56.5–62.2 | | |
| **Years of education (N = 1625)** | | | | | | | **6.87** | **0.032** |
| Less than 8 | 732(46.7) | 44.3–49.2 | 328 (43.3) | 39.8–46.9 | 404 (49.9) | 46.4–53.3 | | |
| 8 years up to 11 | 513(32.7) | 30.5–35.1 | 261 (34.5) | 31.2–37.9 | 252 (31.1) | 28.0–34.4 | | |
| 12 or more | 322 (20.6) | 18.6–22.6 | 168 (22.2) | 19.4–25.3 | 154 (19.0) | 16.5–21.9 | | |
| **Symptoms** | | | | | | | | |
| **Number of symptoms** | | | | | | | 12.36 | 0.006 |
| 0 to 1 symptom | 306 (11.5) | 10.4–12.8 | 170(13.4) | 11.6–15.3 | 136 (9.9) | 8.4–11.6 | | |
| 2 to 4 symptoms | 1404 (53.1) | 51.2–55.0 | 646 (50.7) | 48.0–53.5 | 758(55.2) | 52.6–57.8 | | |
| 5 to 9 symptoms | 860 (32.5) | 30.7–34.3 | 427 (33.5) | 31.0–36.2 | 433 (31.5) | 29.1–34.0 | | |
| 10 or more symptoms | 76 (2.9) | 2.3-3.6 | 30 (2.4) | 1.7-3.4 | 46 (3.4) | 2.5-4.4 | | |
| **Risk factors and comorbidities** | | | | | | | | |
| **Smoking (N = 2777)** | | | | | | | 8.03 | 0.005 |
| No | 2635 (94.9) | 94.0–95.6 | 1305 (96.1) | 94.9–97.0 | 1330 (93.7) | 92.3–94.9 | | |
| Yes | 142 (5.1) | 4.4–6.0 | 53 (3.9) | 3.0–5.1 | 89 (6.3) | 5.1–7.7 | | |
| **Obesity (N = 2770)** | | | | | | | 27.24 | <0.001 |
| No | 2437 (88.0) | 86.7–89.1 | 1235 (91.3) | 89.7–92.7 | 1202 (84.8) | 82.9–86.6 | | |
| Yes | 333 (12) | 10.9–13.3 | 118 (8.7) | 7.3–10.3 | 215 (15.2) | 13.4–17.1 | | |
| **Number of comorbidities (N = 2765)** | | | | | | | 84.74 | <0.001 |
| None | 936 (33.9) | 32.1–35.6 | 554 (41.0) | 38.4–43.7 | 382 (27.0) | 24.8–29.4 | | |
| 1 comorbidity | 850 (30.7) | 29.0–32.5 | 423 (31.3) | 28.9–33.8 | 427 (30.2) | 27.9–32.6 | | |
| 2 or more comorbidities | 979 (35.4) | 33.6–37.2 | 374 (27.7) | 25.4–30.1 | 605 (42.8) | 40.2–45.4 | | |

\* $x^2$: value of the chi-square test for heterogeneity.

Regarding the number of symptoms, most patients presented with 2–4 symptoms (52.8%). However, the number of symptoms was not associated with prolonged hospitalization (p > 0.05). As for risk factors and comorbidities, in the second wave, approximately 4% were smokers, 8.5% were obese, and 28.8% had 2 or more comorbidities. Smoking, obesity, and the number of comorbidities were associated with the duration of hospitalization (p < 0.05).

In the third wave, the majority of individuals were aged up to 59 years (58.5%), were male (54.6%), non-white (57.5%), and had less than 8 years of education (47.8%). Age group was associated with the length of stay (p < 0.001). Individuals aged 60 years or older had a higher frequency of prolonged hospitalization (Table 4).

**Table 3. Sociodemographic and clinical characterization of hospitalizations for COVID-19 according to duration in the 2nd wave, Espírito Santo, 2020-2021.**

| Variables | Sample | | Short (Up to 7 days) | | Long (8 days or more) | | $x^{2*}$ | p-value |
|---|---|---|---|---|---|---|---|---|
| | N (%) | 95% CI | N (%) | 95% CI | N (%) | 95% CI | | |
| **Sociodemographic characteristics** | | | | | | | | |
| **Age range (N = 1821)** | | | | | | | 63.1 | **<0.001** |
| Up to 59 years | 812 (44.6) | 42.3–46.9 | 528 (53.0) | 49.9–56.1 | 284 (34.4) | 31.3–37.7 | | |
| ≥60 | 1009 (55.4) | 53.1–57.7 | 468 (47.0) | 43.9–50.1 | 541 (65.6) | 62.3–68.7 | | |
| **Gender (N = 1821)** | | | | | | | 1.51 | 0.219 |
| Male | 841 (46.2) | 43.9–48.5 | 473 (47.5) | 44.4–50.6 | 368 (44.6) | 41.2–48.0 | | |
| Female | 980 (53.8) | 51.5–56.1 | 523 (52.5) | 49.4–55.6 | 457 (55.4) | 52.0–58.8 | | |
| **Race/color (N = 1454)** | | | | | | | 0.36 | 0.546 |
| White | 681 (46.5) | 43.9–49.0 | 368 (45.8) | 42.4–49.2 | 313 | 43.6–51.2 | | |
| Non-white | 784 (53.5) | 51.0–56.1 | 436 (54.2) | 50.8–57.6 | 348 | 48.8–56.4 | | |
| **Years of education (N = 1106)** | | | | | | | 24.12 | **<0.001** |
| Less than 8 | 479 (43.3) | 40.4–46.3 | 215 (37.1) | 33.3–41.2 | 264 | 45.8–54.4 | | |
| 8 years up to 11 | 393 (35.5) | 32.8–38.4 | 242 (41.8) | 37.8–45.9 | 151 | 24.9–32.7 | | |
| 12 or more | 234 (21.2) | 18.8–23.7 | 122 (21.1) | 17.9–24.6 | 112 | 18.0–25.0 | | |
| **Symptoms** | | | | | | | | |
| **Number of symptoms (N = 1814)** | | | | | | | 3.55 | 0.314 |
| 0 to 1 symptom | 272 (15) | 13.4–16.7 | 163 (16.4) | 14.2–18.9 | 109 (13.3) | 11.1–15.8 | | |
| 2 to 4 symptoms | 958 (52.8) | 50.5–55.1 | 518 (52.2) | 49.1–55.3 | 440 (53.6) | 50.2–57.0 | | |
| 5 to 9 symptoms | 551 (30.4) | 28.3–32.5 | 294 (29.6) | 26.8–32.5 | 257 (31.3) | 28.2–34.6 | | |
| 10 or more symptoms | 33 (1.8) | 1.3-2.5 | 18 (1.8) | 1.1-2.9 | 15 (1.8) | 1.1-3.0 | | |
| **Risk factors and comorbidities** | | | | | | | | |
| **Smoking (N = 1820)** | | | | | | | 12.58 | **<0.001** |
| No | 1754 (96.4) | 95.4–97.1 | 973 (97.8) | 96.7–98.5 | 781 (94.7) | 92.9–96.0 | | |
| Yes | 66 (3.6) | 2.9–4.6 | 22 (2.2) | 1.5–3.3 | 44 (5.3) | 4.0–7.1 | | |
| **Obesity (N = 1818)** | | | | | | | 8.88 | **0.003** |
| No | 1663 (91.5) | 90.1-92.7 | 926 (93.2) | 91.5-94.7 | 737 (89.3) | 87.0-91.3 | | |
| Yes | 155 (8.5) | 7.3–9.9 | 67 (6.8) | 5.3–8.5 | 88 (10.7) | 8.7–13.0 | | |
| **Number of comorbidities (N = 1811)** | | | | | | | 56.51 | **<0.001** |
| None | 829 (45.8) | 43.5–48.1 | 526 (53.0) | 49.9–56.1 | 303 (27.0) | 33.8–40.4 | | |
| 1 comorbidity | 460 (25.4) | 23.4–27.5 | 243 (24.5) | 21.9–27.3 | 217 (26.5) | 23.6–29.6 | | |
| 2 or more comorbidities | 522 (28.8) | 26.8–31.0 | 223 (22.5) | 20.0–25.2 | 299 (36.5) | 33.3–39.9 | | |

\* $x^2$: value of the chi-square test for heterogeneity

Furthermore, most of the sample had 2–4 symptoms (54.5%). The number of symptoms was associated with prolonged hospitalization (p < 0.05). Individuals with 5–9 symptoms had a higher occurrence of prolonged hospitalization than short-term hospitalization (33.3% vs. 26.6%) (p < 0.001). Regarding risk factors and comorbidities, approximately 2% were smokers, 9.3% were obese, and 55.1% had no comorbidity. Smoking, obesity, and the number of comorbidities were associated with the length of stay (p < 0.05), as shown in Table 4.

When analyzing the factors associated with long hospital stays, Table 5 highlights several findings. During the first wave, in the adjusted analysis, the chance of prolonged hospitalization in elderly patients (60 years or older) was 67% higher compared to patients aged 59 years or younger, after adjusting for sex and years of education (OR=1.67; 95%

**Table 4. Sociodemographic and clinical characterization of hospitalizations for COVID-19 according to duration in the 3rd wave, Espírito Santo, 2020-2021.**

| Variables | Sample | | Short (Up to 7 days) | | Long (8 days or more) | | $x^2$* | p-value |
|---|---|---|---|---|---|---|---|---|
| | N (%) | 95% CI | N (%) | 95% CI | N (%) | 95% CI | | |
| **Sociodemographic characteristics** | | | | | | | | |
| **Age range (N=4410)** | | | | | | | 95.3 | **<0.001** |
| Up to 59 years | 2579 (58.5) | 57.0–59.9 | 1493 (65.5) | 63.5–67.4 | 1086 (51.0) | 48.9–53.1 | | |
| ≥60 | 1831 (41.5) | 40.1–43.0 | 787 (34.5) | 32.6-36.5 | 1044 (49.0) | 46.9-51.1 | | |
| **Gender (N=4410)** | | | | | | | 2.1 | 0.148 |
| Male | 2004 (45.4) | 44.0–46.9 | 1060 (46.5) | 44.5–48.5 | 944 (44.3) | 42.2–46.4 | | |
| Female | 2406 (54.6) | 53.1–56.0 | 1220 (53.5) | 51.5–55.5 | 1186 (55.7) | 53.6–57.8 | | |
| **Race/color (N=3472)** | | | | | | | 2.18 | 0.14 |
| White | 1476 (42.5) | 40.9–44.2 | 785 (43.7) | 41.4–46.0 | 691 (41.2) | 38.9–43.6 | | |
| Non-white | 1996 (57.5) | 55.8–59.1 | 1011 (56.3) | 54.0–58.6 | 985 (57.8) | 56.4–61.1 | | |
| **Years of education N=2268)** | | | | | | | 2.88 | 0.237 |
| Less than 8 | 1083 (47.8) | 45.7–49.8 | 530 (48.1) | 45.1–51.0 | 553 (47.5) | 44.6–50.3 | | |
| 8 years up to 11 | 831 (36.6) | 34.7–38.6 | 415 (37.6) | 34.8–40.5 | 416 (35.7) | 33.0–38.5 | | |
| 12 or more | 354 (15.6) | 14.2–17.2 | 158 (14.3) | 12.3–16.5 | 196 (16.8) | 14.8–19.1 | | |
| **Symptoms** | | | | | | | | |
| **Number of symptoms (N=4402)** | | | | | | | 26.92 | **<0.001** |
| 0 to 1 symptom | 608 (13.8) | 12.8–14.9 | 347 (15.2) | 13.8–16.8 | 261 (12.3) | 10.9–13.7 | | |
| 2 to 4 symptoms | 2404 (54.6) | 53.1–56.1 | 1280 (56.3) | 54.2–58.3 | 1124 (52.8) | 50.7–54.9 | | |
| 5 to 9 symptoms | 1313 (29.8) | 28.5–31.2 | 604 (26.6) | 24.8–28.4 | 709 (33.3) | 31.3–35.4 | | |
| 10 or more symptoms | 77 (1.8) | 1.4–2.2 | 43 (1.9) | 1.4-2.5 | 34 (1.6) | 1.1-2.2 | | |
| **Risk factors and comorbidities** | | | | | | | | |
| **Smoking (N=4404)** | | | | | | | 5.45 | **0.02** |
| No | 4311 (97.9) | 97.4–98.3 | 2242 (98.4) | 97.8–98.8 | 2069 (97.4) | 96.6–98.0 | | |
| Yes | 93 (2.1) | 1.7-2.6 | 37 (1.6) | 1.2-2.2 | 56 (2.6) | 2.0-3.4 | | |
| **Obesity (N=4404)** | | | | | | | 54.7 | **<0.001** |
| No | 3996 (90.7) | 89.8–91.6 | 2139 (93.9) | 92.8–94.8 | 1857 (87.4) | 85.9–88.7 | | |
| Yes | 408 (9.3) | 8.4-10.2 | 140 (6.1) | 5.2-7.2 | 268 (12.6) | 11.3-14.1 | | |
| **Number of comorbidities (N=4401)** | | | | | | | 113.6 | **<0.001** |
| None | 2423 (55.1) | 53.6–56.5 | 1415 (62.1) | 60.2–64.1 | 1008 (47.5) | 45.3–49.6 | | |
| 1 comorbidity | 1027 (23.3) | 22.1–24.6 | 493 (21.7) | 20.0–23.4 | 534 (25.1) | 23.3–27.0 | | |
| 2 or more comorbidities | 951 (21.6) | 20.4–22.8 | 368 (16.2) | 14.7–17.7 | 583 (27.4) | 25.6–29.4 | | |

* $x^2$: value of the chi-square test for heterogeneity

CI: 1.35–2.06). Furthermore, individuals with 10 or more symptoms had a 2.03 times higher chance of prolonged hospitalization compared to those with up to 1 symptom (OR=2.03; 95% CI: 1.04–3.94). Obese individuals with 2 or more comorbidities also had a 2.22 times greater chance of prolonged hospitalization compared to non-obese individuals without comorbidities (OR=2.22; 95% CI: 1.71–2.89,).

In the second wave, individuals aged 60 years or older had a 2.04 times higher chance of prolonged hospitalization compared to younger individuals (up to 59 years), controlling for years of education (OR=2.04, 95% CI: 1.58–2.62). Individuals with 8–11 years of education had a 36% lower chance of prolonged hospitalization compared to those with 12 or more years of education, after controlling for potential confounding factors (OR=0.64; 95% CI: 0.46–0.90,. After

**Table 5. Multivariate analysis (adjusted odds ratios using logistic regression model) for the outcome of prolonged hospitalization in COVID-19 hospitalized patients in Espírito Santo, during the first three pandemic waves, from 2020 to 2021.**

| Variables | 1st wave | 2nd wave | 3rd wave |
|---|---|---|---|
| **Sociodemographic variables** | | | |
| **Age group** | | | |
| ≥60 | **1.67***** | **2.04***** | **1.89***** |
| **Sex** | | | |
| Male | 1.2 | – | – |
| **Race/color** | | – | – |
| Non-white | **–** | – | – |
| **Years of education** | | | |
| Less than 8 | 1.16 | 1.06 | – |
| 8 years up to 11 | 1.06 | **0.64**** | – |
| **Clinic variables** | | | |
| **number of symptoms** | | | |
| 2 to 4 symptoms | 1.44 | – | 1.19 |
| 5 to 9 symptoms | 1.21 | – | **1.52***** |
| 10 or more symptoms | **2.03*** | – | 0.91 |
| **Risk factors and comorbidities** | | | |
| **Smoking** | | | |
| Yes | 1.53 | 1.9 | 1.29 |
| **Obesity** | | | |
| Yes | **2.04***** | **1.45**** | **2.3***** |
| **Number of comorbidities** | | | |
| 1 comorbidity | **1.32*** | 1.24 | **1.45***** |
| 2 or more comorbidities | **2.22***** | **1.77***** | **2.04***** |
| (*p<0,05; **p<0,01; ***p<0,001) | | | |

adjustments, individuals with 2 or more comorbidities also had a 77% higher chance of prolonged hospitalization compared to those without any comorbidities (OR=1.77; 95% CI: 1.29–2.41). The number of symptoms was not associated with the duration of hospitalization in the second wave (p>0.05).

Regarding the third wave, after adjustments, individuals aged 60 years or older had an 89% higher chance of prolonged hospitalization compared to individuals aged 59 years or younger (OR=1.89; 95% CI: 1.65–2.17). As for individuals with 5–9 symptoms, they had a 52% higher chance of prolonged hospitalization compared to individuals with up to 1 symptom (OR=1.52; 95% CI: 1.20–1.92). Additionally, obese individuals had a 2.2 times greater chance of prolonged hospitalization compared to non-obese individuals (OR=2.3; 95% CI: 1.78–2.73). When considering potential confounding factors, individuals with one comorbidity and those with 2 or more comorbidities had a 1.45 (95% CI: 1.22–1.72) and 2.0 (95% CI: 1.69–2.45) times higher chance of prolonged hospitalization, respectively, compared to individuals without any comorbidities (p<0.001).

When analyzing the relationship between hospitalization duration and pandemic waves, Table 6 shows that the regression model was statistically significant (p<0.001). Compared to the first wave, the second wave was associated with a 21.0% lower likelihood of prolonged hospitalization, while the third wave showed an 11.0% reduction in this likelihood.

Additionally, using hospitalization duration in days as a continuous dependent variable, the findings indicate that during the second wave there was a significant reduction in the mean length of stay (−0.57 days, p=0.001). This reduction was not observed in the third wave (Table 7).

**Table 6. Association between hospitalization duration for COVID-19 and pandemic waves, 2020–2021.**

| Dependent variable – Length of hospital stay (prolonged hospitalization) | | OR | p-value |
|---|---|---|---|
| Waves | 1st | 1 | – |
| | 2nd | 0.79 | **< 0.001** |
| | 3rd | 0.89 | **0.020** |

Prob χ² (2, 15.09) < 0.001; Pseudo-R² = 2.0%; Omnibus Test (p) = 0.001; Hosmer and Lemeshow test (p) = 0.999; (*) Simple robust regression; OR – Odds Ratio; (0) comparison category; significant if p ≤ 0.050

**Table 7. Association between length of hospital stay for COVID-19 and pandemic waves, 2020–2021.**

| Dependent variable – Length of hospital stay for COVID 19 (Days) | | *B* | Robust standard error | p-value | Trend |
|---|---|---|---|---|---|
| Waves | 1st | 0 | – | – | – |
| | 2nd | −0.57 | 0.17 | **0.001** | Decrease |
| | 3rd | −0.24 | 0.14 | 0.075 | Not significant |

Prob F (2, 9014) = 0.004; R² = 2.4%; Durbin-Watson = 1.97

(*) Simple robust regression; (0) comparison category; significant if p ≤ 0.050

The results, therefore, indicate statistically significant variations in the duration of hospitalizations for COVID-19 across the first three pandemic waves, reflecting gradual changes in the clinical profile and in the public management of care for these patients.

## Discussion

The present study investigated the factors associated with COVID-19 hospitalization duration during across the first three pandemic waves. Prolonged hospitalizations were more frequent during the first wave, a period marked by the introduction of the SARS-CoV-2 virus. However, the most significant risk factors consistently associated with extended hospital stays across all three waves were age group—indicating that older adults (≥60 years) experienced longer hospitalizations compared to younger individuals—along with smoking, obesity, and the number of comorbidities. Notably, in the first wave, prolonged hospitalizations were associated with the number of symptoms. In the second wave, educational attainment was linked to hospitalization duration, with individuals with lower education levels experiencing longer hospital stays. In the third wave, symptom burden was again a key factor associated with prolonged hospitalization.

Recent studies have indicated that among patients with COVID-19, older individuals and those with any comorbidities tend to experienced worse clinical outcomes. In this context, evidence suggest that older adults are more likely to progress to severe disease and have a higher prevalence of comorbidities, contributing to increased disease severity and mortality compared to younger patient groups [18]. Additionally, older adults have shown greater susceptibility to the psychological impact of a pandemic, frequently experiencing emotions such as sadness, anxiety, stress, anger, fear, and distress. This heightened vulnerability is likely due to the uncertainty surrounding treatment outcomes and the potentially fatal course of the disease [19,20].

The increased vulnerability of older adults to prolonged hospitalization due to COVID-19 can be attributed to several molecular and pathophysiological mechanisms. Aging is associated with immunosenescence, characterized by a decline in T and B lymphocyte function, which compromises the immune response against SARS-CoV-2. Additionally, chronic low-grade inflammation, known as inflammaging, leads to increased production of pro-inflammatory cytokines such as IL-6, TNF-α, and IL-1β, exacerbating the cytokine storm frequently observed in severe cases [21,22].

The presence of comorbidities—such as diabetes, hypertension, and cardiovascular diseases, which are more prevalent among older individuals—further aggravates systemic inflammation and endothelial dysfunction, increasing the risk of complications such as thrombosis and multiorgan failure. Lastly, the reduced functional reserve of organs and physiological systems impairs recovery from infection-induced damage, contributing to prolonged hospital stays [21,22].

According to the COVID-19 and *Sustainable Development* report released by the United Nations (UN), skin color and educational attainment are key determinants of COVID-19 fatality rates among Brazilians [23]. The spread of COVID-19 was more pronounced among socioeconomically disadvantaged populations, particularly those with lower levels of education [24]. Thus, the education and race/ethnicity are crucial variables in the pursuit of health equity. owever, these factors are often underreported in official records, leading to gaps and biases in data interpretation [12]. This underscores the essential role of epidemiological surveillance in collecting, validating, and analyzing data to provide an accurate representation of the pandemic's impact [25].

In this study, the number of symptoms presented by patients was associated with a longer duration of hospitalization. The primary symptoms of COVID-19 are fever, cough, dyspnea, myalgia, confusion, headache, odynophagia, rhinorrhea, chest pain, nausea, vomiting [26], anosmia and ageusia [27], conjunctivitis, dermatological manifestations [28], bilateral pneumonia, ground-glass opacity, pneumothorax, and lymphopenia [26]. Although most infected individuals develop mild (40%) or moderate (40%) symptoms, approximately 15% progress to severe disease requiring oxygen support, while around 5% experience a critical form with complications, necessitating hospitalization [29].

A cross-sectional study conducted in Espírito Santo, Brazil found a higher likelihood of hospitalization and mortality among individuals infected with SARS-CoV-2 who were obese and had comorbidities [30], which is consistent with our findings. However, early studies did not initially identify obesity as a significant risk factor for COVID-19 complications [31,32]. As the pandemic progressed and further research emerged, obesity became increasingly recognized as a key factor associated. For instance, a study conducted in China, reported a 142% increase in the risk of developing severe pneumonia among obese individuals compared to those with normal weight [33].

In Mexico, confirmed COVID-19 cases among individuals with obesity had a mortality rate of 13.5% and a higher likelihood of hospitalization, pneumonia, as well as higher rates of ICU admission and intubation compared to to individuals with normal weight [34]. Similarly, in United States, a body mass index (BMI) > 40 kg/m$^2$ has been identified as a significant predictor of hospitalization, second only to advanced age [35].

Obesity has been identified as a significant risk factor for prolonged hospitalization due to COVID-19, driven by multiple molecular and pathophysiological mechanisms described in the literature. Obese individuals often exhibit an impaired immune response, characterized by altered T and B lymphocyte function and an excessive production of pro-inflammatory cytokines such as TNF-α and IL-6. This exacerbated inflammatory response contributes to systemic inflammation and cytokine storm, a critical factor in severe COVID-19 cases. Additionally, obesity is associated with insulin resistance and an increased risk of comorbidities such as hypertension, type 2 diabetes, and cardiovascular diseases, further aggravating disease complications. Excess adipose tissue, particularly in the abdominal region, can impair respiratory function, reducing pulmonary ventilation efficiency and increasing the likelihood of severe respiratory complications. Chronic low-grade inflammation, a hallmark of obesity, also contributes to tissue deterioration and weakens the body's response to viral infections, leading to prolonged recovery times and extended hospitalization [36,37].

COVID-19 infection is associated with an increased risk of prolonged hospitalization, particularly among patients with comorbidities such as cardiovascular diseases, chronic obstructive pulmonary disease (COPD), chronic kidney disease, and malignancies. Understanding the molecular and pathophysiological mechanisms underlying this elevated risk is essential for effective clinical management. Patients with preexisting cardiovascular diseases are at a higher risk of severe complications from COVID-19. The SARS-CoV-2 virus binds to the angiotensin-converting enzyme 2 (ACE2) receptor, which is expressed in cardiac and endothelial cells. This interaction can lead to endothelial dysfunction, inflammation, and

thrombosis, exacerbating preexisting cardiac conditions and increasing the likelihood of complications such as myocardial infarction, myocarditis, and arrhythmias [38,39].

Similarly, patients with COPD face an elevated risk of severe respiratory infections due to compromised lung function and chronic inflammation. COVID-19 infection can further exacerbate pulmonary inflammation and respiratory dysfunction, often necessitating ventilatory support and prolonging hospitalization [38,40]. Additionally, chronic kidney disease can complicate the clinical course and delay recovery. Patients with this condition are more susceptible to acute kidney injury due to factors such as hypoxia, systemic inflammation, and endothelial dysfunction [41].

Patients with malignancies are at an increased risk of severe COVID-19 due to immunosuppression associated with both cancer treatment and the disease itself. The exacerbated inflammatory response and immune dysfunction characteristic of these patients can contribute to a more severe and prolonged clinical course [38,42].

These mechanisms are often aggravated by systemic inflammation and the cytokine storm induced by SARS-CoV-2 infection, which can result in multiorgan dysfunction and thromboembolic complications [40,41]. Understanding these molecular and pathophysiological processes is crucial for developing clinical management strategies aimed at reducing hospitalization duration and improving outcomes in COVID-19 patients with underlying comorbidities.

Elderly individuals face a markedly elevated risk of severe COVID-19–related physical complications, which can partially explain their consistently prolonged hospital stays. Advancing age is associated with a gradual decline in immune competence (immunosenescence), reduced respiratory reserve, and impaired mucociliary clearance, leading to higher susceptibility to pulmonary complications such as acute respiratory distress syndrome (ARDS) and respiratory failure. Additionally, older patients more frequently develop extrapulmonary complications—including acute kidney injury, acute myocardial injury, and acute liver injury—that drive clinical deterioration and prolong hospitalization duration [43,44].

Moreover, aging is closely linked with an increased burden of chronic comorbidities—such as hypertension, cardiovascular disease, chronic kidney disease, diabetes mellitus, and chronic lung disease—that potentiate the severity of SARS-CoV-2 infection and complicate recovery trajectories. Data from the CDC indicate that adults aged ≥65 years account for the majority of COVID-19 hospitalizations, ICU admissions, and in-hospital deaths, with most carrying multiple underlying conditions [45]. These intersecting physiological susceptibilities and comorbid states contribute substantively to longer hospital stays among elderly patients and underscore the necessity of tailored clinical and public health interventions.

In light of the findings of this study, it is important to highlight that in Brazil, during the first wave, a lack of knowledge regarding COVID-19 management, combined with the absence of standardized protocols and delayed diagnoses, led to prolonged hospitalizations, particularly among older adults and patients with multiple comorbidities. In the second wave, social inequalities became more evident, with individuals of lower educational attainment facing a higher risk of prolonged hospital stays due to limited access to health information and precarious living conditions, underscoring the need for inclusive public health strategies. By the third wave, despite progress in vaccination, the number of symptoms remained a key determinant of prolonged hospitalizations, reflecting the impact of aggressive variants and suboptimal clinical management. These findings reinforce the importance of epidemiological surveillance and the continuous adaptation of clinical protocols to evolving pandemic challenges [14,42].

The primary limitation of this study is its retrospective design, which relies on preexisting secondary database. Underreporting and a relatively high proportion of missing data in key variables—particularly those related to patient hospitalization—represent notable constraints, leading to the exclusion of a significant portion of the sample. Additionally, clinical and comorbidity data were primarily based on self-reported information at hospital admission, which may contributed to an underestimating of the true strength of the associations with clinical prognosis. Furthermore, reported data on mortality, discharge status, and hospital length of stay may be subject to revisions as the database undergoes further refinement by the Health Surveillance Service of the municipalities across Espírito Santo state. Despite these limitations, the findings of this study align with global literature and can contribute to epidemiological analyses of COVID-19 trends throughout the

pandemic waves, extending beyond the southeastern Brazilian context. Future studies incorporating external validation of these results are recommended to strengthen the robustness of these findings.

## Conclusion

This study identified clinical and sociodemographic factors associated with prolonged hospital stays among COVID-19 patients in a southeastern state of Brazil. Advanced age, low educational attainment, presence of symptoms, obesity, and comorbidities were significantly associated with extended hospitalization. The pandemic evolved differently across successive waves, highlighting the importance of understanding these patterns to inform public health strategies. Such insights can guide the Health Surveillance Service in developing targeted interventions to mitigate viral transmission and, most importantly, implement health promotion and disease prevention measures in vulnerable populations, who face an increased risk of hospitalization and prolonged stays. Investments in active surveillance and case monitoring, particularly in primary healthcare settings, are crucial for effective disease management and control.

## Supporting information

**S1 File. Statistical analysis Juliana [30abr23].**
(PDF)

## Acknowledgments

We would like to thank the Instituto Federal do Espírito Santo (Ifes) and Secretaria de Estado da Saúde do Espírito Santo—SESA/ES for its support of this research, as well as for its authorization and provision of secondary data.

## Author contributions

**Conceptualization:** Juliana Rodrigues Tovar Garbin, Franciéle Marabotti Costa Leite, Luís Carlos Lopes-Júnior.

**Data curation:** Juliana Rodrigues Tovar Garbin, Franciéle Marabotti Costa Leite, Ana Paula Brioschi dos Santos, Larissa Soares Dell'Antonio, Cristiano Soares da Silva Dell'Antonio, Luís Carlos Lopes-Júnior.

**Formal analysis:** Juliana Rodrigues Tovar Garbin, Franciéle Marabotti Costa Leite, Luís Carlos Lopes-Júnior.

**Investigation:** Juliana Rodrigues Tovar Garbin, Franciéle Marabotti Costa Leite.

**Methodology:** Juliana Rodrigues Tovar Garbin, Franciéle Marabotti Costa Leite, Luís Carlos Lopes-Júnior.

**Project administration:** Franciéle Marabotti Costa Leite, Luís Carlos Lopes-Júnior.

**Supervision:** Franciéle Marabotti Costa Leite, Luís Carlos Lopes-Júnior.

**Validation:** Juliana Rodrigues Tovar Garbin, Franciéle Marabotti Costa Leite, Ana Paula Brioschi dos Santos, Larissa Soares Dell'Antonio, Cristiano Soares da Silva Dell'Antonio, Luís Carlos Lopes-Júnior.

**Visualization:** Juliana Rodrigues Tovar Garbin, Franciéle Marabotti Costa Leite, Ana Paula Brioschi dos Santos, Larissa Soares Dell'Antonio, Cristiano Soares da Silva Dell'Antonio, Luís Carlos Lopes-Júnior.

**Writing – original draft:** Juliana Rodrigues Tovar Garbin, Franciéle Marabotti Costa Leite, Ana Paula Brioschi dos Santos, Larissa Soares Dell'Antonio, Cristiano Soares da Silva Dell'Antonio, Luís Carlos Lopes-Júnior.

**Writing – review & editing:** Juliana Rodrigues Tovar Garbin, Franciéle Marabotti Costa Leite, Ana Paula Brioschi dos Santos, Larissa Soares Dell'Antonio, Cristiano Soares da Silva Dell'Antonio, Luís Carlos Lopes-Júnior.

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
