## [Decision Letter · Decision Letter 0]

13 Nov 2024

PONE-D-24-05103

Factors associated with prolonged hospitalizations for COVID-19 during the first three waves of the pandemic: evidence from a Southeastern state of Brazil

PLOS ONE

Dear Dr. Lopes-Júnior,

Thank you for submitting your manuscript to PLOS ONE. After careful consideration, we feel that it has merit but does not fully meet PLOS ONE’s publication criteria as it currently stands. Therefore, we invite you to submit a revised version of the manuscript that addresses the points raised during the review process.

We look forward to receiving your revised manuscript.

Kind regards,

André Ricardo Ribas Freitas

Academic Editor

PLOS ONE

Journal Requirements:

2. We note that your Data Availability Statement is currently as follows: “All relevant data are within the manuscript and in Supporting Information files.”

Please confirm at this time whether or not your submission contains all raw data required to replicate the results of your study. Authors must share the “minimal data set” for their submission. PLOS defines the minimal data set to consist of the data required to replicate all study findings reported in the article, as well as related metadata and methods (https://journals.plos.org/plosone/s/data-availability#loc-minimal-data-set-definition). For example, authors should submit the following data: - The values behind the means, standard deviations and other measures reported; - The values used to build graphs; - The points extracted from images for analysis. Authors do not need to submit their entire data set if only a portion of the data was used in the reported study. If your submission does not contain these data, please either upload them as Supporting Information files or deposit them to a stable, public repository and provide us with the relevant URLs, DOIs, or accession numbers. For a list of recommended repositories, please see https://journals.plos.org/plosone/s/recommended-repositories. If there are ethical or legal restrictions on sharing a de-identified data set, please explain them in detail (e.g., data contain potentially sensitive information, data are owned by a third-party organization, etc.) and who has imposed them (e.g., an ethics committee). Please also provide contact information for a data access committee, ethics committee, or other institutional body to which data requests may be sent. If data are owned by a third party, please indicate how others may request data access.

Reviewers' comments:

Reviewer's Responses to Questions

**Comments to the Author**

1. Is the manuscript technically sound, and do the data support the conclusions?

Reviewer #1: Yes

Reviewer #2: Yes

2. Has the statistical analysis been performed appropriately and rigorously? 

Reviewer #1: Yes

Reviewer #2: Yes

3. Have the authors made all data underlying the findings in their manuscript fully available?

Reviewer #1: Yes

Reviewer #2: Yes

4. Is the manuscript presented in an intelligible fashion and written in standard English?

Reviewer #1: Yes

Reviewer #2: Yes

5. Review Comments to the Author

Reviewer #1: Dear Authors

The current manuscript is outstanding but needs to improve in some ways. Even so, there are some questions.

Introduction

The authors of the current manuscript could briefly describe complications of prolonged hospitalization due to COVID-19, such as long-term COVID-19.

Materials and Methods

In the second paragraph, what do the authors mean about 19?... in the second paragraph, what do authors mean about 19...

The 138 adopted inclusion criterion was confirmation of the diagnosis of COVID-19, with 6 139 laboratory confirmations by RT-PCR and hospitalization in Espírito Santo due to this 140 diagnosis.

Measures

The authors must report the types and places of hospitalized patients with COVID-19, such as wards or intensive care units.

Could the authors report outcomes, such as the need for mechanical ventilation, dialysis, or even mortality, to compare the different waves of the COVID-19 pandemic?

Results

Was there any statistical difference in the distribution of hospitalizations between the three waves?

Authors need to write better about significant comorbidities, comparing each wave to help understand any changes in comorbidities across these waves.

Again, were the authors able to determine outcomes related to length of hospital stay?

Was there a longer length of hospital stay in intensive care units in which wave of the COVID-19 pandemic?

Discussion

The authors could describe molecular and pathophysiological mechanisms evident in the literature and explain why some comorbidities are related to a higher risk of prolonged hospitalization due to COVID-19. For example, in the elderly, is a reduced innate immunity predisposed to more severe SARS-CoV-2 infection?

It is crucial to underscore the potential for a prolonged hospitalization period related to outcomes, as this highlights the severity of the disease and the significant need for hospital support, including oxygen and ventilatory support or vasoactive drugs. As the authors have noted, "an estimated 15% of patients may develop severe symptoms that necessitate oxygen support, and about 5% may progress to the critical form of the disease with complications necessitating hospital support".

Furthermore, the authors have carried out their research with the utmost ethical standards, ensuring the integrity and reliability of the data. This commitment to ethical research practices should instill confidence in the audience about the validity of the findings.

Reviewer #2: Thank you for investing time and efforts studying this topic. The topic is very important that should augment global efforts to improve preparedness for upcoming pandemic. Please consider the following:

1- Define the level of support the state had offered in the acknowledgement section

2- Revise English

3- Improve the introduction section; shed light on Brazilian COVID-19 experience and Brazilian pandemic mitigation strategies

4- Reflect on the rationale of the study.

5- Revise the discussion section. Link the results to the pandemic management protocol

6. PLOS authors have the option to publish the peer review history of their article (what does this mean? ). If published, this will include your full peer review and any attached files.

**Do you want your identity to be public for this peer review?** For information about this choice, including consent withdrawal, please see our Privacy Policy .

Reviewer #1: **Yes: ** Miguel Angelo Goes

Reviewer #2: **Yes: ** WEAM Mohammed BANJAR

---

## [Author Response · Author response to Decision Letter 1]

22 Mar 2025

Vitória, February 26th 2025

Response Letter to the Reviewers

Dear Editors and Reviewers,

We sincerely appreciate the opportunity to revise and improve our manuscript based on the insightful comments and constructive suggestions provided by the reviewers. We believe that these recommendations have significantly enhanced the quality and clarity of our study. Below, we provide a detailed point-by-point response to each comment, indicating the revisions made in the manuscript.

Decision: Minor Revision

Reviewer 1:

a) Introduction: The authors of the current manuscript could briefly describe complications of prolonged hospitalization due to COVID-19, such as long-term COVID-19.

Response: Ok. Done! The following sentence was added: "It is important to highlight that patients with severe manifestations, such as acute respiratory distress syndrome (ARDS) and those requiring intensive care unit (ICU) admission, are more likely to develop long-term sequelae due to the persistence of symptoms, commonly referred to as long COVID-19."

b) Materials and Methods: In the second paragraph, what do the authors mean about 19?

Response: OK. This was a typographical error. The number was removed.

c) Measures: The authors must report the types and places of hospitalized patients with COVID-19, such as wards or intensive care units.

Response: Both general ward and intensive care unit (ICU) hospitalizations were considered. The following sentence was added: "Hospitalizations in both general wards and intensive care units (ICUs) were taken into account."

d) Could the authors report outcomes, such as the need for mechanical ventilation, dialysis, or even mortality, to compare the different waves of the COVID-19 pandemic?

Response: This study relied on secondary data from state notification records. Unfortunately, data on mechanical ventilation and dialysis were not included in these records. Since our objective was to analyze risk factors associated with hospitalization, mortality was not analyzed. However, we have previously published a study on survival analysis (Garbin, J.R.T. et al., Int. J. Environ. Res. Public Health 2022, 19, 8709. https://doi.org/10.3390/ijerph19148709). Future studies may explore this further.

e) Results: Was there any statistical difference in the distribution of hospitalizations between the three waves?

Response: Yes. Table 1 indicates a statistically significant difference in hospitalization distribution, as shown by the 95% confidence interval, particularly between the first and second waves and the first and third waves. Additional information, including the p-value, has been included in line 149. We also published similar results in another study (Garbin, J.R.T. et al., Sci Rep 2024, 14, 5777. https://doi.org/10.1038/s41598-024-56289-7).

f) Authors need to write better about significant comorbidities, comparing each wave to help understand any changes in comorbidities across these waves.

Response: Thank you for this relevant comment. The text was adjusted for improved clarity.

g) Again, were the authors able to determine outcomes related to the length of hospital stay?

Response: Yes, we observed that prolonged hospitalizations were more frequent during the first wave. The most important risk factors consistently associated with long-term hospitalizations were age (with patients aged 60 years or older more likely to experience prolonged stays), smoking, obesity, and comorbidities. Specifically, in the first wave, prolonged hospitalizations were associated with the number of symptoms; in the second wave, lower educational attainment was associated with longer stays; and in the third wave, symptom burden was again a key factor.

h) Was there a longer length of hospital stay in intensive care units in which wave of the COVID-19 pandemic?

Response: Our study did not separately analyze ICU hospitalizations by wave. Both general ward and ICU hospitalizations were included in our analysis, as our objective was to examine factors associated with short-term and prolonged hospitalization for COVID-19 across the three pandemic waves, regardless of hospitalization type.

i) The authors could describe molecular and pathophysiological mechanisms evident in the literature and explain why some comorbidities are related to a higher risk of prolonged hospitalization due to COVID-19. For example, in the elderly, is reduced innate immunity predisposed to more severe SARS-CoV-2 infection?

Response: Yes, we have added literature describing molecular mechanisms in the discussion, as suggested. Thank you!

Reviewer 2:

1. Define the level of support the state had offered in the acknowledgment section.

Response: The government of Espírito Santo, through the State Health Department, authorized and provided secondary data for this study. This information has been added to the acknowledgment section.

2. Revise English.

Response: The entire manuscript has been revised for English language improvement, as recommended.

3. Improve the introduction section; shed light on the Brazilian COVID-19 experience and Brazilian pandemic mitigation strategies.

Response: We have added a paragraph addressing this aspect.

4. Reflect on the rationale of the study.

Response: We have rewritten the justification to enhance clarity and better reflect the manuscript's content.

5. Revise the discussion section. Link the results to the pandemic management protocol.

Response: The discussion section has been revised, and the results have been linked to Brazil’s pandemic management protocols. A paragraph was added at the end of the discussion.

We sincerely appreciate all the comments and suggestions, which have substantially improved our manuscript. Thank you for your time and consideration.

Best regards,

The Authors

---

## [Decision Letter · Decision Letter 1]

20 Jul 2025

PONE-D-24-05103R1Factors associated with prolonged hospitalizations for COVID-19 during the first three waves of the pandemic: evidence from a Southeastern state of BrazilPLOS ONE

Dear Dr.  Lopes-Júnior,

Thank you for submitting your manuscript to PLOS ONE. After careful consideration, we feel that it has merit but does not fully meet PLOS ONE’s publication criteria as it currently stands. Therefore, we invite you to submit a revised version of the manuscript that addresses the points raised during the review process.

We look forward to receiving your revised manuscript.

Kind regards,

Ivan Filipe de Almeida Lopes Fernandes, Ph.D.

Academic Editor

PLOS ONE

Journal Requirements:

Additional Editor Comments:

The manuscript “Factors associated with prolonged hospitalizations for COVID-19 during the first three waves of the pandemic: evidence from a Southeastern state of Brazil” presents a retrospective observational study analyzing factors associated with short and prolonged hospitalizations due to COVID-19. Using data from Espírito Santo state in Brazil, the manuscript classified hospitalizations as short or prolonged based on a 7-day threshold. They conducted bivariate analyses using chi-square tests and employed logistic regression models to explore associations between clinical and sociodemographic factors and hospitalization duration across the first three pandemic waves. The study’s main finding is that prolonged hospitalizations were more frequent in the first wave. In contrast, shorter stays predominated in the second and third waves, with older age, obesity, symptom burden, and comorbidities associated with prolonged hospitalization.

Given that I have assumed editorial responsibility after prior rounds of peer review, and considering the manuscript has already addressed comments from previous reviewers, I will request only a limited set of fundamental revisions before the manuscript can be considered for acceptance. If these changes are fully addressed, the manuscript will be approved for publication.

1) Empirical Testing of the Main Hypothesis: the manuscript’s argument regarding differences in the proportion of prolonged hospitalizations across the three waves is presented emphatically. Still, it lacks formal statistical testing to support the conclusion. I request the authors to perform two additional analyses:

First, a logistic regression model using hospitalization duration (short vs. long) as the dependent variable and the pandemic wave as the key independent variable for the full dataset, composed of the three waves. A second model using hospitalization duration in days as a continuous dependent variable, preferably applying an appropriate regression model to compare the mean duration across waves. These analyses are essential to substantiate the claim that the profile of hospitalization duration significantly changed across waves. The conclusions about differences between waves should only be presented after conducting and reporting these tests.

2) Throughout the manuscript, the authors redundantly report the confidence level (95% CI) and the alpha level (5%) when describing statistical results. It is unnecessary to state both, as a 95% confidence interval inherently implies an alpha of 5%. Please standardize the reporting and remove redundant mentions of the alpha level where confidence intervals are already presented.

3) Definition of Long vs. Short Hospitalizations: the use of 7 days as the threshold to define short and prolonged hospitalizations needs further justification. The current explanation that it corresponds to the median of the second wave is insufficient. Please elaborate on why the second wave median was chosen as the cutoff and discuss its clinical relevance. Additionally, as a robustness check, I recommend testing alternative thresholds (e.g., 6 days or 10 days) and reporting whether the main associations remain consistent.

4) Standardization of Regression Tables: The presentation of regression results should follow a more conventional format. Specifically, remove the reference category rows (“Ref.”) as a separate line in the table. Present only the Odds Ratios (OR) and their standard errors. Confidence intervals and p-values should be omitted from the table body, and p-values should be indicated using conventional asterisks (*p<0.05; **p<0.01; ***p<0.001). Consider eliminating the crude (unadjusted) models from the main regression tables and moving them to a statistical appendix if necessary. In the main text, I recommend presenting only the adjusted models, using a clearer layout to improve readability.

MINNOR COMMENTS

a) Page 2, line 62:

From: “Subsequently, the Odds/Odds Ratio and their respective 95% confidence intervals”

To: “Subsequently, the Odds Ratios and their respective 95% confidence intervals”

b) Page 3, line 81:

Please clarify what is meant by “distinct patterns,” as the results across the waves do not appear markedly discrepant. Consider rephrasing to reflect better the observed differences, which seem subtle rather than pronounced.

c) Page 5, line 138:

From: The adopted inclusion criterion was confirmation of the diagnosis for COVID-19 19 with laboratory confirmation by RT-PCR ...

To: The adopted inclusion criterion was confirmation of the diagnosis for COVID-19 with laboratory confirmation by RT-PCR ...

d) Page 14, line 242

From: the likelihood of prolonged hospitalization in elderly patients

To: the chance of prolonged hospitalization in elderly patients

d)Page 20, line 292-300:

Consider expanding the discussion regarding the vulnerabilities of older adults. While the current text highlights their susceptibility to psychological effects, it is important to emphasize that older adults are also at greater risk for a wide range of physical health complications. Beyond mental health issues, elderly individuals often experience other deteriorating body functions that may exacerbate COVID-19 outcomes. Deepening this analysis would provide a more comprehensive understanding of why age is consistently associated with prolonged hospitalizations and worse clinical outcomes.

I look forward to reviewing the revised version incorporating these essential modifications.

Regards,

Reviewers' comments:

Reviewer's Responses to Questions

**Comments to the Author**

1. If the authors have adequately addressed your comments raised in a previous round of review and you feel that this manuscript is now acceptable for publication, you may indicate that here to bypass the “Comments to the Author” section, enter your conflict of interest statement in the “Confidential to Editor” section, and submit your "Accept" recommendation.

Reviewer #1: All comments have been addressed

Reviewer #2: All comments have been addressed

2. Is the manuscript technically sound, and do the data support the conclusions?

Reviewer #1: Yes

Reviewer #2: Yes

3. Has the statistical analysis been performed appropriately and rigorously? 

Reviewer #1: Yes

Reviewer #2: Yes

4. Have the authors made all data underlying the findings in their manuscript fully available?

Reviewer #1: Yes

Reviewer #2: Yes

5. Is the manuscript presented in an intelligible fashion and written in standard English?

Reviewer #1: Yes

Reviewer #2: Yes

6. Review Comments to the Author

Reviewer #1: Dear authors,

You have conducted association research that aligns with the local context at the time. Additionally, you have addressed all the questions raised by the reviewer and improved the manuscript accordingly. There are no further questions.

Reviewer #2: Thank you for taking the time to address reviewers comments and concerns. The manuscript had been substantially improved.

7. PLOS authors have the option to publish the peer review history of their article (what does this mean? ). If published, this will include your full peer review and any attached files.

**Do you want your identity to be public for this peer review?** For information about this choice, including consent withdrawal, please see our Privacy Policy .

Reviewer #1: **Yes: ** Miguel Angelo Goes, MD, PhD, FASN

Reviewer #2: **Yes: ** WEAM BANJAR

---

## [Author Response · Author response to Decision Letter 2]

12 Aug 2025

Vitória, ES, on August 12, 2025

Manuscript title: Factors associated with prolonged hospitalizations for COVID-19 during the first three waves of the pandemic: evidence from a Southeastern state of Brazil

Response Letter to the Associate Editor and Editor-in-Chief of PLOS One,

Editor Comments:

The manuscript “Factors associated with prolonged hospitalizations for COVID-19 during the first three waves of the pandemic: evidence from a Southeastern state of Brazil” presents a retrospective observational study analyzing factors associated with short and prolonged hospitalizations due to COVID-19. Using data from Espírito Santo state in Brazil, the manuscript classified hospitalizations as short or prolonged based on a 7-day threshold. They conducted bivariate analyses using chi-square tests and employed logistic regression models to explore associations between clinical and sociodemographic factors and hospitalization duration across the first three pandemic waves. The study’s main finding is that prolonged hospitalizations were more frequent in the first wave. In contrast, shorter stays predominated in the second and third waves, with older age, obesity, symptom burden, and comorbidities associated with prolonged hospitalization.

Given that I have assumed editorial responsibility after prior rounds of peer review, and considering the manuscript has already addressed comments from previous reviewers, I will request only a limited set of fundamental revisions before the manuscript can be considered for acceptance. If these changes are fully addressed, the manuscript will be approved for publication.

Response: We sincerely appreciate the opportunity to revise and improve our manuscript in response to the insightful comments and constructive suggestions provided. We fully agree with the points raised, and we have incorporated all the recommended changes into the manuscript as advised. We are confident that these revisions have enhanced the clarity, robustness, and overall quality of our study.

Below, we provide a detailed point-by-point response to each editorial comment, indicating the specific modifications made in the revised version.

Decision: Minor Revision

1.Empirical Testing of the Main Hypothesis: the manuscript’s argument regarding differences in the proportion of prolonged hospitalizations across the three waves is presented emphatically. Still, it lacks formal statistical testing to support the conclusion. I request the authors to perform two additional analyses:

First, a logistic regression model using hospitalization duration (short vs. long) as the dependent variable and the pandemic wave as the key independent variable for the full dataset, composed of the three waves. A second model using hospitalization duration in days as a continuous dependent variable, preferably applying an appropriate regression model to compare the mean duration across waves. These analyses are essential to substantiate the claim that the profile of hospitalization duration significantly changed across waves. The conclusions about differences between waves should only be presented after conducting and reporting these tests.

Response: As recommended, we performed the additional statistical analyses to substantiate our conclusions regarding differences in hospitalization duration across the three pandemic waves. Specifically, we applied a simple robust regression model with robust standard errors to account for potential violations of the homoscedasticity assumption. This approach provides estimates that are more resilient to the presence of outliers and does not require the residuals to follow a normal distribution, which is an important premise in this context.

The assumption of serial autocorrelation was evaluated using the Durbin–Watson test. Multicollinearity was not assessed, as only one independent variable was included in the model. The regression evaluated the association between COVID-19 hospitalization duration and the pandemic wave of occurrence, with the alpha level for statistical significance set at 5%.

For the logistic regression model, the analysis was statistically significant (p < 0.001), allowing for valid inferences. The model’s explanatory power (pseudo R²) was 2.0%, indicating that it explained 2.0% of the variation in the dependent variable. The Omnibus test was also significant (p = 0.001), confirming that the independent variable influenced the outcome. Furthermore, the Hosmer–Lemeshow test indicated no rejection of the null hypothesis that the model fit the data well (p = 0.999). Compared to the first wave, the second wave was associated with a 21.0% lower likelihood of prolonged hospitalization (≥8 days) relative to short hospitalization (≤7 days), while the third wave showed an 11.0% reduction (Table 6).

For the second model, which used hospitalization duration in days as a continuous dependent variable, the regression was also statistically significant (p = 0.004), enabling valid inferences. The explanatory power (pseudo R²) was 2.4%, meaning the model explained 2.4% of the variation in the dependent variable. The Durbin–Watson statistic was close to 2, indicating no evidence of serial autocorrelation. In this model, the second wave of COVID-19 showed a significant trend toward a reduced mean length of stay compared with the first wave (−0.57 days, p = 0.001), whereas the third wave was not statistically significant, indicating that the mean hospitalization duration was not influenced by this wave (Table 7).

2) Throughout the manuscript, the authors redundantly report the confidence level (95% CI) and the alpha level (5%) when describing statistical results. It is unnecessary to state both, as a 95% confidence interval inherently implies an alpha of 5%. Please standardize the reporting and remove redundant mentions of the alpha level where confidence intervals are already presented.

Response: All redundant mentions of the alpha level were removed in sentences where the 95% confidence interval was already reported. The statistical reporting has been standardized throughout the manuscript to ensure clarity and avoid unnecessary repetition.

3) Definition of Long vs. Short Hospitalizations: the use of 7 days as the threshold to define short and prolonged hospitalizations needs further justification. The current explanation that it corresponds to the median of the second wave is insufficient. Please elaborate on why the second wave median was chosen as the cutoff and discuss its clinical relevance. Additionally, as a robustness check, I recommend testing alternative thresholds (e.g., 6 days or 10 days) and reporting whether the main associations remain consistent.

Response: We appreciate this observation. In this study, the classification of short-term and prolonged hospitalizations across the three pandemic waves was based on a 7-day cutoff, corresponding to the median hospital stay observed during the second wave. This parameter was similar to the study by Wu et al. (2020), which also conducted its analyses according to the median hospitalization time, using univariable and multivariable logistic regression methods to identify risk factors associated with long-term hospitalization in patients with COVID-19 [16].

Wu Y, Hou B, Liu J, Chen Y, Zhong P. Risk factors associated with long-term hospitalization in patients with COVID-19: A single-centered, retrospective study. Front Med (Lausanne). 2020;7:315. doi:10.3389/fmed.2020.00315.

In addition, a Brazilian study published in PLOS ONE (de Andrade et al., 2020) reported that the length of hospital stay for COVID-19 patients admitted to the Brazilian Unified Health System (SUS) ranged from less than 24 hours to 114 days, with an average of 6.9 days, which corroborates our findings [17].

de Andrade CLT, Pereira CCdA, Martins M, Lima SML, Portela MC. COVID-19 hospitalizations in Brazil’s Unified Health System (SUS). PLoS One. 2020;15(12):e0243126. doi:10.1371/journal.pone.0243126.

Accordingly, the Methods section was updated to read: “Using the median length of stay as a threshold has been reported in the literature as a method to assess hospital course in homogeneous groups with distinct clinical outcomes [16]. Furthermore, the choice of a 7-day cutoff is consistent with national data indicating an average hospital stay of 6.9 days among patients admitted to public hospitals within the Brazilian Unified Health System (SUS), reinforcing the clinical representativeness of this threshold for our study population [17].”

4) Standardization of Regression Tables: The presentation of regression results should follow a more conventional format. Specifically, remove the reference category rows (“Ref.”) as a separate line in the table. Present only the Odds Ratios (OR) and their standard errors. Confidence intervals and p-values should be omitted from the table body, and p-values should be indicated using conventional asterisks (*p<0.05; **p<0.01; ***p<0.001). Consider eliminating the crude (unadjusted) models from the main regression tables and moving them to a statistical appendix if necessary. In the main text, I recommend presenting only the adjusted models, using a clearer layout to improve readability.

Response: The regression tables have been reformatted according to the recommendation. The reference category rows (“Ref.”) were removed as separate lines, and only the Odds Ratios (OR) and their standard errors are now presented. Confidence intervals and p-values have been omitted from the table body, and p-values are indicated using the conventional asterisk notation (*p < 0.05; **p < 0.01; ***p < 0.001). Additionally, the crude (unadjusted) models have been moved to the statistical appendix, and only the adjusted models are presented in the main text to improve clarity and readability.

MINNOR COMMENTS

a) Page 2, line 62:

From: “Subsequently, the Odds/Odds Ratio and their respective 95% confidence intervals”

To: “Subsequently, the Odds Ratios and their respective 95% confidence intervals”

Response:: The text has been corrected as suggested.

b) Page 3, line 81:

Please clarify what is meant by “distinct patterns,” as the results across the waves do not appear markedly discrepant. Consider rephrasing to reflect better the observed differences, which seem subtle rather than pronounced.

Response: The sentence has been revised for clarity and accuracy. It now reads: “In conclusion, we observed variations in hospitalization patterns across pandemic waves, although the differences between them were subtle. These variations reflect gradual changes in risk factors associated with prolonged hospital stays.”

c) Page 5, line 138:

From: The adopted inclusion criterion was confirmation of the diagnosis for COVID-19 19 with laboratory confirmation by RT-PCR ...

To: The adopted inclusion criterion was confirmation of the diagnosis for COVID-19 with laboratory confirmation by RT-PCR ...

Response: The text has been corrected as suggested.

d) Page 14, line 242

From: the likelihood of prolonged hospitalization in elderly patients

To: the chance of prolonged hospitalization in elderly patients

Response: The text has been corrected as suggested.

d)Page 20, line 292-300:

Consider expanding the discussion regarding the vulnerabilities of older adults. While the current text highlights their susceptibility to psychological effects, it is important to emphasize that older adults are also at greater risk for a wide range of physical health complications. Beyond mental health issues, elderly individuals often experience other deteriorating body functions that may exacerbate COVID-19 outcomes. Deepening this analysis would provide a more comprehensive understanding of why age is consistently associated with prolonged hospitalizations and worse clinical outcomes.

Response: We have addressed this comment by adding two new paragraphs at the end of the Discussion section, elaborating on the broader vulnerabilities of older adults beyond psychological effects. The revised text emphasizes their increased risk for multiple physical health complications, the impact of comorbidities, and age-related physiological decline that may exacerbate COVID-19 outcomes. Three additional references were included to support this expanded discussion.

Yours Sincerely,

The authors

---

## [Editor Report · Decision Letter 2]

27 Aug 2025

Factors associated with prolonged hospitalizations for COVID-19 during the first three waves of the pandemic: evidence from a Southeastern state of Brazil

PONE-D-24-05103R2

Dear Dr. Lopes-Júnior,

We’re pleased to inform you that your manuscript has been judged scientifically suitable for publication and will be formally accepted for publication once it meets all outstanding technical requirements.

Kind regards,

Ivan Filipe de Almeida Lopes Fernandes, Ph.D.

Academic Editor

PLOS ONE
---

## [Editor Report · Acceptance letter]

PONE-D-24-05103R2

PLOS ONE

Dear Dr. Lopes-Júnior,

I'm pleased to inform you that your manuscript has been deemed suitable for publication in PLOS ONE. Congratulations! Your manuscript is now being handed over to our production team.

Kind regards,

on behalf of

Dr. Ivan Filipe de Almeida Lopes Fernandes

Academic Editor

PLOS ONE